# Angiographic Features and Clinical Outcomes of Balloon Uncrossable Lesions during Chronic Total Occlusion Percutaneous Coronary Intervention

**DOI:** 10.3390/jpm13030515

**Published:** 2023-03-13

**Authors:** Judit Karacsonyi, Spyridon Kostantinis, Bahadir Simsek, Athanasios Rempakos, Salman S. Allana, Khaldoon Alaswad, Oleg Krestyaninov, Jaikirshan Khatri, Paul Poommipanit, Farouc A. Jaffer, James Choi, Mitul Patel, Sevket Gorgulu, Michalis Koutouzis, Ioannis Tsiafoutis, Abdul M. Sheikh, Ahmed ElGuindy, Basem Elbarouni, Taral Patel, Brian Jefferson, Jason R. Wollmuth, Robert Yeh, Dimitrios Karmpaliotis, Ajay J. Kirtane, Margaret B. McEntegart, Amirali Masoumi, Rhian Davies, Bavana V. Rangan, Olga C. Mastrodemos, Darshan Doshi, Yader Sandoval, Mir B. Basir, Michael S. Megaly, Imre Ungi, Nidal Abi Rafeh, Omer Goktekin, Emmanouil S. Brilakis

**Affiliations:** 1Center for Coronary Artery Disease, Abbott Northwestern Hospital, Minneapolis Heart Institute and Minneapolis Heart Institute Foundation, Minneapolis, MN 55407, USA; 2Department of Cardiology, Henry Ford Hospital, Detroit, MI 48202, USA; 3Meshalkin Novosibirsk Research Institute, Novosibirsk 630055, Russia; 4Department of Cardiovascular Medicine, Cleveland Clinic, Cleveland, OH 44195, USA; 5Cardiology, Case Western Reserve University, University Hospitals, Cleveland, OH 44610, USA; 6Department of Cardiology, Massachusetts General Hospital, Boston, MA 02114, USA; 7Department of Cardiology, Baylor Heart and Vascular Hospital, Dallas, TX 75226, USA; 8Cardiovascular Institute, University of California San Diego, VA San Diego Healthcare System, La Jolla, CA 92037, USA; 9Department of Cardiology, Biruni University School of Medicine, Istanbul 34295, Turkey; 10First Cardiology Department Athens, Red Cross Hospital of Athens, Athens 11526, Greece; 11Interventional Cardiology Department, Wellstar Health System, Marietta, GA 30141, USA; 12Aswan Heart Centre, Department of Cardiology, Magdi Yacoub Foundation, Aswan 4271185, Egypt; 13Department of Internal Medicine, St. Boniface General Hospital, Winnipeg, MB R2H 2A6, Canada; 14Interventional Cardiology, Tristar Centennial Medical Center, Nashville, TN 37203, USA; 15Interventional Cardiology, Providence Heart institute, Portland, OR 97213, USA; 16Division of Cardiology, Beth Israel Deaconess Medical Center, Boston, MA 02215, USA; 17Interventional Cardiology, Morristown Medical Center, Gagnon Cardiovascular Institute, Morristown, NJ 07960, USA; 18Division of Cardiology, Columbia University, New York, NY 10032, USA; 19Interventional Cardiology, WellSpan York Hospital, York, PA 17403, USA; 20Division of Invasive Cardiology, Department of Internal Medicine and Cardiology Center, University of Szeged, 6725 Szeged, Hungary; 21Cardiology, North Oaks Health System, Hammond, LA 70403, USA; 22Department of Cardiology, Memorial Bahcelievler Hospital, Istanbul 34676, Turkey

**Keywords:** percutaneous coronary intervention, chronic total occlusion, balloon uncrossable

## Abstract

**Background:** Balloon uncrossable lesions are defined as lesions that cannot be crossed with a balloon after successful guidewire crossing. **Methods:** We analyzed the association between balloon uncrossable lesions and procedural outcomes of 8671 chronic total occlusions (CTOs) percutaneous coronary interventions (PCIs) performed between 2012 and 2022 at 41 centers. **Results:** The prevalence of balloon uncrossable lesions was 9.2%. The mean patient age was 64.2 ± 10 years and 80% were men. Patients with balloon uncrossable lesions were older (67.3 ± 9 vs. 63.9 ± 10, *p* < 0.001) and more likely to have prior coronary artery bypass graft surgery (40% vs. 25%, *p* < 0.001) and diabetes mellitus (50% vs. 42%, *p* < 0.001) compared with patients who had balloon crossable lesions. In-stent restenosis (23% vs. 16%. *p* < 0.001), moderate/severe calcification (68% vs. 40%, *p* < 0.001), and moderate/severe proximal vessel tortuosity (36% vs. 25%, *p* < 0.001) were more common in balloon uncrossable lesions. Procedure time (132 (90, 197) vs. 109 (71, 160) min, *p* < 0.001) was longer and the air kerma radiation dose (2.55 (1.41, 4.23) vs. 1.97 (1.10, 3.40) min, *p* < 0.001) was higher in balloon uncrossable lesions, while these lesions displayed lower technical (91% vs. 99%, *p* < 0.001) and procedural (88% vs. 96%, *p* < 0.001) success rates and higher major adverse cardiac event (MACE) rates (3.14% vs. 1.49%, *p* < 0.001). Several techniques were required for balloon uncrossable lesions. **Conclusion:** In a contemporary, multicenter registry, 9.2% of the successfully crossed CTOs were initially balloon uncrossable. Balloon uncrossable lesions exhibited lower technical and procedural success rates and a higher risk of complications compared with balloon crossable lesions.

## 1. Introduction

Balloon uncrossable lesions are defined as lesions that cannot be crossed with a balloon after successful guidewire crossing and confirmation of the guidewire position in the true lumen [1,2,3]. In a prior publication from the PROGRESS-CTO registry, the prevalence of balloon uncrossable lesions was 9%. These challenging lesions often required the use of multiple complex treatment modalities and were associated with lower technical and procedural success rates [2].

Several treatment strategies are available for treating balloon uncrossable lesions and can be broadly categorized into (a) plaque modification techniques, and (b) techniques that increase guide catheter support. An algorithmic approach to balloon uncrossable lesions usually starts with the use of small, low-profile balloons, followed by techniques that increase guide catheter support, and various plaque modification strategies, such as the use of microcatheters, atherectomy, laser, and extraplaque lesion modification [3]. We examined the contemporary clinical outcomes of balloon uncrossable CTO PCI.

## 2. Materials and Methods

### 2.1. Study Population

We analyzed the baseline clinical and angiographic characteristics and procedural outcomes of 8671 CTO PCIs with successful guidewire crossing performed between 2012 and 2022, at 41 centers. Data collection was recorded in a dedicated online database (PROGRESS-CTO: Prospective Global Registry for the Study of Chronic Total Occlusion Intervention; Clinicaltrials.gov identifier: NCT02061436) [2,4,5,6,7]. Study data were collected and managed using REDCap (Research Electronic Data Capture) electronic data capture tools hosted at the Minneapolis Heart Institute Foundation [8,9]. The study was approved by the institutional review board of each site.

### 2.2. Definitions

Coronary CTOs were defined as coronary lesions with Thrombolysis in Myocardial Infarction (TIMI), grade 0 flow of at least 3 months duration. Estimation of the duration of occlusion was clinical, based on the first onset of angina, prior history of myocardial infarction (MI) in the target vessel territory, or comparison with a prior angiogram. Calcification was assessed by angiography as mild (spots), moderate (involving ≤ 50% of the reference lesion diameter), or severe (involving > 50% of the reference lesion diameter). Moderate proximal vessel tortuosity was defined as the presence of at least 2 bends > 70° or 1 bend > 90°, and severe tortuosity as 2 bends > 90° or 1 bend > 120° in the CTO vessel. A retrograde procedure was an attempt to cross the lesion through a collateral vessel or bypass graft supplying the target vessel distal to the lesion; otherwise, the intervention was classified as an antegrade-only procedure. Antegrade dissection/re-entry was defined as antegrade PCI during which a guidewire was intentionally introduced into the subintimal space proximal to the lesion, or re-entry into the distal true lumen was attempted after intentional or inadvertent subintimal guidewire crossing [4].

### 2.3. Outcomes

Technical success was defined as successful CTO revascularization with the achievement of <30% residual diameter stenosis within the treated segment and restoration of TIMI grade 3 antegrade flow. Procedural success was defined as the achievement of technical success without any in-hospital major adverse cardiac event (MACE), which was defined as any of the following events prior to hospital discharge: death, MI, recurrent symptoms requiring urgent repeat target-vessel revascularization (TVR) with PCI, or coronary artery bypass graft (CABG) surgery, tamponade requiring either pericardiocentesis or surgery, and stroke. MI was defined using the Third Universal Definition of Myocardial Infarction (type 4a MI) [4,6,7,10]. The Japanese CTO (J-CTO) score was calculated as described by Morino et al. [11], the PROGRESS-CTO score was as described by Christopoulos et al. [12], and the PROGRESS-CTO MACE score was as described by Simsek et al. [13].

### 2.4. Statistics

Categorical variables were expressed as percentages and compared using Pearson’s Chi-square test. Continuous variables were presented as mean ± standard deviation or median (interquartile range (IQR)), unless otherwise specified, and were compared using the student’s *t*-test for normally distributed variables and the Kruskal–Wallis test for non-parametric variables, as appropriate. The variables associated with technical success and periprocedural MACE were examined using univariable logistic regression; thereafter, logistic multivariable regression was performed, and the variables with *p* values over 0.1 were removed from the model. All other statistical analyses were performed using JMP, version 13.0 (SAS Institute). A two-sided *p*-value of < 0.05 was considered statistically significant [4,5,6].

## 3. Results

Among cases with successful guidewire crossing the prevalence of balloon uncrossable lesions was 9.2%. The mean patient age was 64.2 ± 10 years, 80% were men, 43% had diabetes mellitus, approximately half had a prior MI (45%), and approximately one-third had prior artery coronary bypass surgery (27%).

Table 1 represents the baseline clinical characteristics of the study patients classified according to the presence of balloon uncrossable lesions. Patients with balloon uncrossable lesions were older (67.3 ± 9 vs. 63.9 ± 10, *p* < 0.001), more likely to have had prior coronary artery bypass graft surgery (40% vs. 25%, *p* < 0.001), diabetes mellitus (50% vs. 42%, *p* < 0.001), and peripheral arterial disease (17% vs. 13%, *p* = 0.004) compared with patients who had balloon crossable lesions (Table 1).

The right coronary artery (52%) was the most common target vessel, followed by the left anterior descending coronary artery (27%), and the left circumflex (19%). Overall, the most common successful crossing strategy was antegrade wire escalation (66%), followed by the retrograde approach (20%), and antegrade dissection and re-entry (14%). The mean J-CTO score was 2.27 ± 1.26, the mean PROGRESS-CTO score was 1.13 ± 0.98, and the mean PROGRESS-CTO MACE score was 2.46 ± 1.63. Moderate or severe calcification was present in 42% and in-stent restenosis in 16% of the cases. The baseline angiographic and procedural characteristics of the target lesions, classified according to whether they were uncrossable or not, are demonstrated in Table 1. Moderate or severe calcification (68% vs. 40%, *p* < 0.001) and proximal vessel tortuosity (36% vs. 25%, *p* < 0.001) were more common in balloon uncrossable lesions, which were also more complex with higher mean J-CTO (2.58 ± 1.19 vs. 2.23 ± 1.28, *p* < 0.001), PROGRESS-CTO (1.30 ± 1.02 vs. 1.11 ± 0.97, *p* < 0.001), and PROGRESS MACE (2.66 ± 1.54 vs. 2.44 ± 1.64, *p* < 0.001) scores compared with PCI of balloon crossable CTOs.

Procedural outcomes and techniques are shown in Table 2, the Graphical Abstract, and Figure 1. In cases where successful guidewire crossing was achieved, the overall technical and procedural success rates were 98% and 96%, respectively, and the incidence of in-hospital MACE was 1.64%. Balloon uncrossable lesions had lower technical (91% vs. 99%, *p* < 0.001) and procedural (88% vs. 96%, *p* < 0.001) success and higher incidence of major adverse cardiac events (3.14% vs. 1.49%, *p* < 0.001). Procedure time (132 (90, 197) vs. 109 (71, 160) min, *p* < 0.001) was longer and the air kerma radiation dose (2.55 (1.41, 4.23) vs. 1.97 (1.10, 3.40) min, *p* < 0.001) and contrast volume (210 (150, 300) vs. 200 (145, 280) mL, *p* = 0.001, Figure 1) were higher in balloon uncrossable lesions. Several techniques were used in balloon uncrossable lesions, such as guide catheter extensions in 267 cases (34%), grenadoplasty in 198 cases (25%), rotational atherectomy in 179 cases (23%), and laser in 140 cases (18%, Figure 2). Rotational atherectomy combined with laser atherectomy was used in 31 cases (4%), while rotational atherectomy, together with orbital atherectomy, was used in 3 cases (0.4%). Intravascular lithotripsy was used in 5 cases (0.6%).

On multivariable analyses moderate/severe calcification, longer occlusion length, and balloon uncrossable lesions were associated with lower technical success, while bigger vessel diameter and the presence of interventional collaterals were associated with higher technical success (Figure 3A). Balloon uncrossable lesions were also associated with higher MACE on multivariable analyses along with the presence of interventional collaterals, cerebrovascular disease, and moderate/severe calcification (Figure 3B).

## 4. Discussion

The key findings of our study are that: (a) in a contemporary multicenter registry, 9.2% of CTOs that were successfully crossed with a wire were balloon uncrossable; (b) balloon uncrossable lesions had lower technical and procedural success and higher risk of complications compared with balloon crossable lesions; and (c) balloon uncrossable lesions often required use of advanced plaque modification and increased support techniques.

In a prior publication from the PROGRESS-CTO registry, the prevalence of balloon uncrossable lesions following successful guidewire crossing was 9.0% [2], whereas the prevalence of balloon uncrossable or balloon undilatable lesions was 15.5% [4], and the prevalence of balloon undilatable lesions was 8.5% [14]. The prevalence was similar in the present study (9.2%). In this study, balloon uncrossable lesions were more complex, with a higher prevalence of moderate or severe calcification, proximal vessel tortuosity, and higher J-CTO and PROGRESS-CTO scores. The high prevalence of balloon uncrossable CTOs also could be explained by new guidewires enabling antegrade intraplaque crossing, which could contribute to more difficulties with microcatheter, and balloon crossing compared with extraplaque crossing.

In our study perforation rates were higher in the balloon uncrossable group compared with the balloon crossable group (6.04% vs. 3.42%, *p* < 0.001). Moreover, pericardiocentesis was also more common in the balloon uncrossable group (1.76% vs. 0.69%, *p* = 0.001). In a prior study from the PROGRESS-CTO registry, the pericardiocentesis rates were numerically higher but not statistically different, which could be due to smaller sample size (1.6% vs. 0.6%, *p* = 0.388) [2]. In a study examining GuideLiner use in balloon uncrossable CTO PCI, only one distal wire perforation occurred in 28 CTO PCI cases, which was treated conservatively without any hemodynamic compromise and no pericardial effusion on echocardiography [15]. A prior study examining rotational atherectomy in balloon uncrossable CTOs did not report any perforations associated with rotational atherectomy [16], neither did the study by Fernandez et al. with balloon uncrossable CTOs and laser atherectomy [17], although these studies had modest sample sizes. The cause of the increased rates of perforations in balloon uncrossable lesions could be explained by higher angiographic complexity (higher J-CTO and PROGRESS-CTO scores, calcification, and proximal vessel tortuosity), as well as need for more complex treatment modalities, which are associated with a higher risk of complications.

Several strategies are available to deal with balloon uncrossable lesions and in general, can be categorized into (a) plaque modification techniques, and (b) techniques that increase guide catheter support [18]. Attempted crossing with low-profile balloons (e.g., 1–1.5 mm diameter), grenadoplasty/balloon assisted microdissection (BAM) [19], and improved guide support with a guide extension [15] or anchor balloon are usually tried first [3]. Other options include rotational [16], orbital or laser atherectomy [18,20], and extraplaque techniques to modify the uncrossable lesion through the extraplaque balloon crush technique or by tracking around the uncrossable plaque, through the less resistant extraplaque space and re-entering the lumen distally [21,22]. Combinations of the various plaque modification techniques can be used if required, such as laser-assisted orbital or rotational atherectomy [18]. The combined use of rotational atherectomy and excimer LaSER is called the “RASER” technique and consists of the upfront use of laser atherectomy followed by rotational atherectomy in heavily calcified lesions after the failure of Rotawire delivery [3,23]. An advantage of laser atherectomy is that it can be performed over any standard 0.014-inch guidewire (although it should be performed with caution over polymer-jacketed guidewires due to the risk of “melting” the polymer). After successful crossing, further modification can be performed with balloons (non-compliant, high-pressure, scoring, cutting), atherectomy devices, or intravascular lithotripsy [18,24]. Image guidance with intravascular ultrasound is valuable after dilatation of initially balloon-uncrossable lesions.

An algorithmic approach to balloon uncrossable lesions usually starts with the use of low-profile balloons, followed by improved guide catheter support, the use of microcatheters, wire cutting or puncture techniques, atherectomy, laser, and extraplaque techniques. Sequential and simultaneous application of these techniques can result in the successful treatment of balloon uncrossable lesions [3,25,26].

There is limited data on the techniques that are the most successful in treating balloon uncrossable lesions. In our study orbital atherectomy, rotational atherectomy, and laser were associated with the highest technical success (Figure 2).

In a single-center study of 290 lesions from 288 cases, with uncrossable lesions treated with rotational or orbital atherectomy, intravascular ultrasound analyses showed that the lesions were not always severely calcified (CTOs were excluded). The interaction of lesion morphology (continuous long and large arcs of calcium) and vessel geometry (bend in the vessel or ostial lesion location) affected lesion crossability [27]. In a prior study from the PROGRESS-CTO registry, which examined the use of atherectomy during chronic total CTO PCI, atherectomy was used in 51 cases (1.4%) as a bailout strategy for “balloon uncrossable” or/and “balloon undilatable” lesions. The cases with “balloon uncrossable” and “balloon undilatable” lesions, where atherectomy was used, had higher technical success rates (92% vs. 79%, *p* = 0.032) and procedural (90% vs. 79%, *p* = 0.046) success rates compared with similar lesions not treated with atherectomy. MACE rates were similar (7% vs. 4%, *p* = 0.422) [28].

The BLIMP study randomized 126 patients with an uncrossable lesion to treatment with the Blimp balloon (Interventional Medical Device Solutions—IMDS, Roden, Netherlands) or low-profile balloon, and found no difference in the first attempt to cross (48% vs. 45%, respectively; *p* = 0.761). After placement of a guide extension, the overall successful lesion crossing was 80% in the BLIMP group compared to 76% in the low-profile balloon group (*p* = 0.327) [29]. In a retrospective study by Ye et al., the efficacy and safety of the BAM technique were assessed in 24 balloon uncrossable CTOs with the Sapphire^®^ II 1.0 mm balloon (OrbusNeich, Hong Kong, China). The technical success rate was 75% (18/24) for the lesions successfully treated with BAM, with a total technical success rate of 92% (22/24; when BAM failed, 2 patients were successfully treated with laser and 2 with rotational atherectomy) [30].

Fernandez et al. assessed the use of laser atherectomy in 58 cases of balloon failure in a single center study in the United Kingdom, 16 of whom had balloon uncrossable CTOs, with a procedural success of 87.5% and 2% incidence of complications (in 2 cases laser was combined with rotational atherectomy). In the same cohort, the laser alone was applied successfully in two balloon undilatable CTO cases but with one Ellis class I perforation [4,17]. The Laser Veterans Affairs (LAVA) study, examining laser use in the veteran population at three US centers undergoing PCI, found balloon uncrossable lesions to be the most common indication for laser (43.8%) associated with 87.8% technical, and 83.7% procedural success rates [4,24]. The LEONARDO (Early outcome of high energy Laser (Excimer) facilitated coronary angioplasty ON hARD and complex calcified and balloOn-resistant coronary lesions) study examined 80 patients with 100 lesions in 4 Italian centers treated with laser atherectomy and described a 93.7% success rate without any complications (perforation, major side branch occlusion, spasm, no-reflow phenomenon, dissection, and acute vessel closure) [4,31].

Another technique for treating balloon and microcatheter uncrossable CTOs is the Carlino and guide-extension Carlino technique, which uses hydraulic disruption by contrast injection via either the microcatheter or guide catheter extension wedged against the uncrossable proximal cap or occlusive segment [32], as well as intentional subintimal dissection and reentry to “go around” the recalcitrant lesion. After the wire entered the extraplaque space, it can re-enter into the distal true lumen with different re-entry techniques, the calcific lesion can be “crushed” with a balloon over the extraplaque wire [22], or distal to the CTO to anchor the true lumen guidewire and allow balloon crossing (“subintimal distal anchor technique”) [3,33].

## 5. Study Limitations

Limitations of our study are that the data is observational, there was no clinical event adjudication or core laboratory analyses, and all procedures were performed at high-volume, experienced CTO PCI centers, limiting the generalizability of our findings to centers with more limited experience.

## 6. Conclusions

In a contemporary multicenter registry, 9.2% of successful wires across CTOs were balloon uncrossable. Balloon uncrossable lesions had lower technical and procedural success and higher complication rates compared to balloon crossable lesions and often required the use of advanced plaque modification and support techniques.

## Figures and Tables

**Figure 1 jpm-13-00515-f001:**
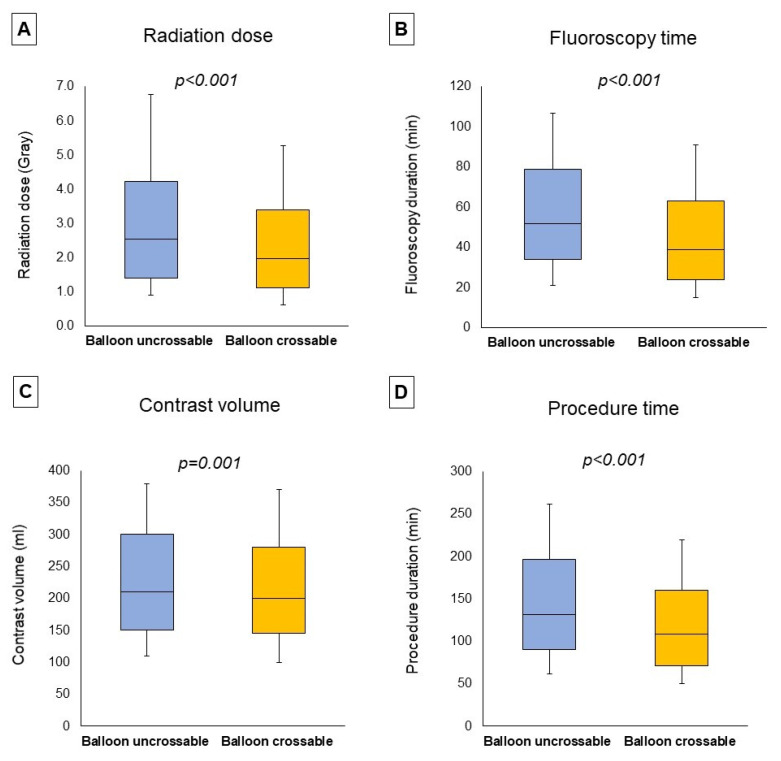
**Procedural characteristics.** The boxplots represent the median radiation dose (**A**), median fluoroscopy time (**B**), median contrast volume (**C**), and median procedure time (**D**) classified according to the presence of balloon uncrossable lesions, showing (from **top** to **bottom**): maximum, Q3, median, Q1, minimum.

**Figure 2 jpm-13-00515-f002:**
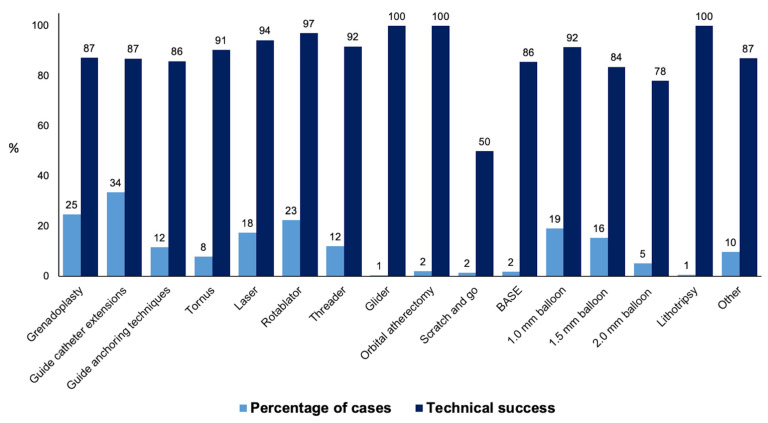
Frequency and technical success rates of the techniques used for balloon uncrossable lesions (BASE: balloon-assisted subintimal entry).

**Figure 3 jpm-13-00515-f003:**
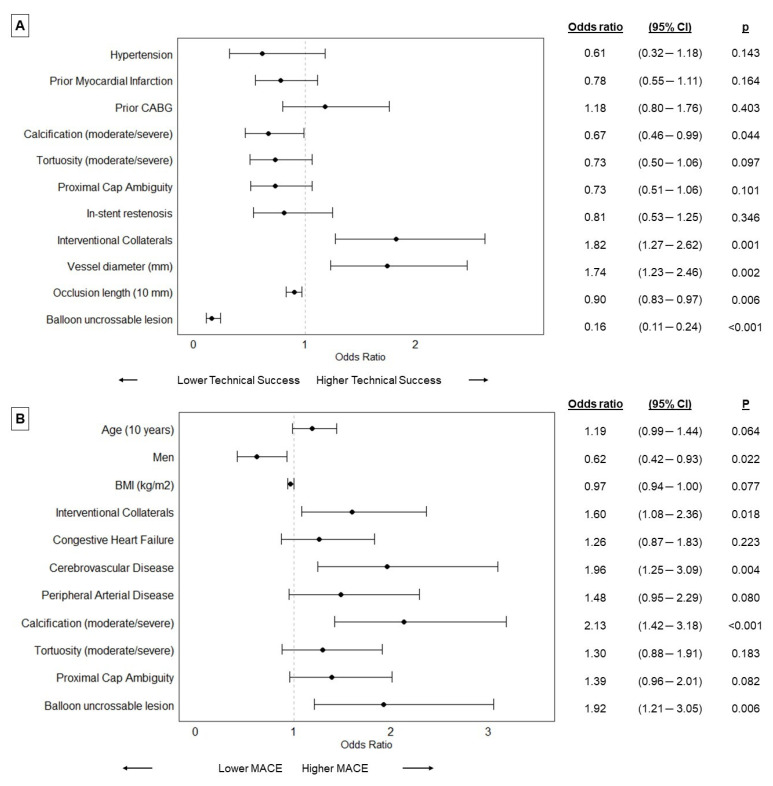
Forest plots representing the results of the multivariable analyses on technical success (**A**) and major cardiac adverse events (MACE, (**B**)) (Dots represent odds ratios, BMI: body mass index; CABG: coronary artery bypass grafting; CI: confidence interval; MACE: major cardiac adverse events).

**Table 1 jpm-13-00515-t001:** Baseline clinical, angiographic, and technical characteristics of study patients with balloon uncrossable and balloon crossable lesions.

Variable	Balloon Uncrossable (n = 795)	Balloon Crossable (n = 7876)	*p* Value
Age (years) *	67.3 ± 9	63.9 ± 10	<0.001
Men	656 (84.8%)	6099 (79.9%)	0.006
BMI (kg/m^2^) *	30.0 ± 6	30.3 ± 6	0.103
Diabetes Mellitus	379 (50.1%)	3075 (41.8%)	<0.001
Hypertension	692 (90.6%)	6539 (87.9%)	0.030
Dyslipidemia	690 (90.8%)	6233 (84.0%)	<0.001
Smoking (current)	165 (20.8%)	1897 (24.1%)	0.036
LVEF (%) *	50 ± 12	50 ± 13	0.670
Family History of CAD	213 (32.9%)	2068 (32.1%)	0.684
Congestive Heart Failure	226 (30.6%)	2014 (28.0%)	0.131
Prior Myocardial Infarction	330 (46.0%)	3136 (44.3%)	0.386
Prior CABG	303 (40.0%)	1866 (25.4%)	<0.001
Prior CVD	81 (10.8%)	730 (10.1%)	0.504
Prior PVD	123 (16.5%)	928 (12.8%)	0.004
Clinical presentation			
▪ Stable angina	477 (63.5%)	4887 (66.4%)	0.001
▪ Unstable angina	127 (16.9%)	1158 (15.7%)	
▪ NSTEMI	85 (11.3%)	578 (7.9%)	
▪ STEMI	3 (0.4%)	110 (1.5%)	
▪ Non-ischemic symptoms	20 (2.7%)	157 (2.1%)	
▪ No symptoms	39 (5.2%)	470 (6.4%)	
Baseline creatinine (mg/dL) †	1.0 (0.9, 1.2)	1.0 (0.9, 1.2)	0.646
CTO Target Vessel			
▪ RCA	435 (56.1%)	3988 (51.6%)	<0.001
▪ LAD	156 (20.1%)	2147 (27.8%)	
▪ LCX	168 (21.7%)	1424 (18.4%)	
▪ SVG	0 (0%)	11 (0.1%)	
▪ LM	4 (0.5%)	38 (0.5%)	
▪ Other	13 (1.7%)	122 (1.6%)	
J-CTO score *	2.58 ± 1.19	2.23 ± 1.28	<0.001
PROGRESS-CTO score *	1.30 ± 1.02	1.11 ± 0.97	<0.001
PROGRESS-CTO MACE score	2.66 ± 1.54	2.44 ± 1.64	<0.001
Calcification (moderate/severe)	544 (68.4%)	3105 (39.4%)	<0.001
Proximal vessel tortuosity (moderate/severe)	289 (36.4%)	1992 (25.3%)	<0.001
Proximal cap ambiguity	210 (27.6%)	2360 (31.6%)	0.025
In-stent restenosis	177 (22.9%)	1185 (15.6%)	<0.001
Side branch at the proximal cap	388 (51.6%)	3965 (54.5%)	0.123
Blunt/no stump, %	379 (47.7%)	4002 (50.8%)	0.092
Vessel diameter (mm) †	3.0 (2.5, 3.0)	3.0 (2.5, 3.0)	0.033
Occlusion length (mm) †	25 (15, 40)	25 (15, 40)	0.060
Number of stents used *	2.3 ± 1.1	2.3 ± 1.1	0.234

BMI: body mass index, LVEF: left ventricular ejection fraction; CAD: coronary artery disease; CABG: coronary artery bypass grafting; CVD: cerebrovascular disease; PVD: peripheral vascular disease; CTO: chronic total occlusion; RCA: right coronary artery, LAD: left descending coronary artery, LCX: left circumflex coronary artery; LM: left main coronary artery; SVG: saphenous vein graft; J-CTO: Japan CTO score; PROGRESS-CTO score: Prospective Global Registry for the Study of Chronic Total Occlusion intervention score; MACE: major adverse cardiac events. *: mean ± standard deviation; †: median (interquartile ranges).

**Table 2 jpm-13-00515-t002:** Procedural characteristics and outcomes of study patients, classified according to the presence of balloon uncrossable lesions.

Variable	Balloon Uncrossable (n = 795)	Balloon Crossable (n = 7876)	*p* Value
Successful Crossing Strategy			
▪ Antegrade wiring	571 (72.0%)	5141 (65.6%)	<0.001
▪ Retrograde	118 (14.9%)	1621 (20.7%)	
▪ ADR	104 (13.1%)	1070 (13.7%)	
First Crossing Strategy			
▪ Antegrade wiring	712 (89.7%)	6706 (85.6%)	0.006
▪ Retrograde	62 (7.8%)	891 (11.4%)	
▪ ADR	20 (2.5%)	237 (3.0%)	
Retrograde crossing strategy	174 (21.9%)	2187 (27.8%)	<0.001
ADR crossing strategy	143 (18.0%)	1393 (17.7%)	0.832
Technical Success	724 (91.1%)	7761 (98.5%)	<0.001
Procedural Success	697 (87.7%)	7585 (96.3%)	<0.001
Procedural time (min) †	132 (90, 197)	109 (71, 160)	<0.001
Fluoroscopy time (min) †	52 (34, 79)	39 (24, 63)	<0.001
Air kerma radiation dose (Gray) †	2.55 (1.41, 4.23)	1.97 (1.10, 3.40)	<0.001
Contrast volume (mL) †	210 (150, 300)	200 (145, 280)	0.001
MACE	25 (3.14%)	117 (1.49%)	<0.001
Death	4 (0.50%)	26 (0.33%)	0.428
Acute MI	7 (0.88%)	31 (0.39%)	0.048
Re-PCI	2 (0.25%)	13 (0.17%)	0.576
Stroke	0 (0.13%)	13 (0.17%)	0.793
Emergency CABG	0 (0%)	1 (0.01%)	0.751
Pericardiocentesis	14 (1.76%)	54 (0.69%)	0.001
Perforation	48 (6.04%)	269 (3.42%)	<0.001
Dissection/Thrombus of Donor Artery	4 (0.50%)	51 (0.65%)	0.625
Aortocoronary Dissection	1 (0.13%)	22 (0.28%)	0.422

MACE: major cardiac adverse events; MI: myocardial infarction; PCI: percutaneous coronary intervention; CABG: coronary artery bypass grafting. †: median (interquartile ranges).

## Data Availability

The authors elect not to share research data.

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
