# Peer review of "Angiographic Features and Clinical Outcomes of Balloon Uncrossable Lesions during Chronic Total Occlusion Percutaneous Coronary Intervention"

_jpm, 2023, doi:10.3390/jpm13030515_

Round 1

Reviewer 1 Report

Congratulation to the authors for the new analysis of the PROGRESS-CTO registry regarding the treatment of the balloon uncrossable lesions.

The present analysis is in line with two before publications of the research group on this topic (referenced 2,10). However, two other publications 

(Xenogiannis I, Karmpaliotis D, Alaswad K, Jaffer FA, Yeh RW, Patel M, Mahmud E, Choi JW, Burke MN, Doing AH, Dattilo P, Toma C, Smith AJC, Uretsky B, Krestyaninov O, Khelimskii D, Holper E, Potluri S, Wyman RM, Kandzari DE, Garcia S, Koutouzis M, Tsiafoutis I, Khatri JJ, Jaber W, Samady H, Jefferson BK, Patel T, Moses JW, Lembo NJ, Parikh M, Kirtane AJ, Ali ZA, Doshi D, Tajti P, Rangan BV, Abdullah S, Banerjee S, Brilakis ES. Usefulness of Atherectomy in Chronic Total Occlusion Interventions (from the PROGRESS-CTO Registry). Am J Cardiol. 2019 May 1;123(9):1422-1428. doi: 10.1016/j.amjcard.2019.01.054. Epub 2019 Feb 11. PMID: 30798947.)

Simsek B, Kostantinis S, Karacsonyi J, Alaswad K, Karmpaliotis D, Masoumi A, Jaffer FA, Doshi D, Khatri J, Poommipanit P, Gorgulu S, Abi Rafeh N, Goktekin O, Krestyaninov O, Davies R, ElGuindy A, Jefferson BK, Patel TN, Patel M, Chandwaney RH, Mastrodemos OC, Rangan BV, Brilakis ES. Prevalence and outcomes of balloon undilatable chronic total occlusions: Insights from the PROGRESS-CTO. Int J Cardiol. 2022 Sep 1;362:42-46. doi: 10.1016/j.ijcard.2022.04.057. Epub 2022 Apr 26. PMID: 35483480.) 

are not mentioned in the manuscript and it is not clear what was the difference between the aims of the present work and the before one. The present analysis includes more cases, but the interval of the collection time is not specified against the before one (between 2012 and 2022). I suggest investigating the recent tendencies in the database according to the new results. 

E.g. in the present manuscript, the numbers of the rotational (23%: 1994 cases) and the orbital atherectomies (2%: 173 cases) have increased significantly compared to the before data. It should be worth giving data about the rate of combined usage of the different types atherotomies. I am curious the opinion of the authors the preference and the strategy for applying the two methods. 

Author Response

                                                                                                            March 2nd, 2023

Prof. Dr. David Alan Rizzieri

Editor-in-Chief: Journal of Personalized Medicine

Senior Vice President and Director, Novant Health Cancer Institute, Winston-Salem, NC USA

Professor of Medicine Chief, Section of Hematologic Malignancies, Associate Director for Clinical Research, Division of Hematologic Malignancies and Cellular Therapy Duke Cancer Institute, Durham, NC, USA

Dear Professor Rizzieri,

Thank you very much for reviewing our manuscript entitled Percutaneous Coronary Interventions of Balloon Uncrossable Chronic Total Occlusions”. We greatly appreciated the reviewers’ comments and revised the manuscript accordingly, as follows:

Reviewer #1

Congratulation to the authors for the new analysis of the PROGRESS-CTO registry regarding the treatment of the balloon uncrossable lesions. The present analysis is in line with two before publications of the research group on this topic (referenced 2,10). However, two other publications (Xenogiannis I, Karmpaliotis D, Alaswad K, Jaffer FA, Yeh RW, Patel M, Mahmud E, Choi JW, Burke MN, Doing AH, Dattilo P, Toma C, Smith AJC, Uretsky B, Krestyaninov O, Khelimskii D, Holper E, Potluri S, Wyman RM, Kandzari DE, Garcia S, Koutouzis M, Tsiafoutis I, Khatri JJ, Jaber W, Samady H, Jefferson BK, Patel T, Moses JW, Lembo NJ, Parikh M, Kirtane AJ, Ali ZA, Doshi D, Tajti P, Rangan BV, Abdullah S, Banerjee S, Brilakis ES. Usefulness of Atherectomy in Chronic Total Occlusion Interventions (from the PROGRESS-CTO Registry). Am J Cardiol. 2019 May 1;123(9):1422-1428. doi: 10.1016/j.amjcard.2019.01.054. Epub 2019 Feb 11. PMID: 30798947. and Simsek B, Kostantinis S, Karacsonyi J, Alaswad K, Karmpaliotis D, Masoumi A, Jaffer FA, Doshi D, Khatri J, Poommipanit P, Gorgulu S, Abi Rafeh N, Goktekin O, Krestyaninov O, Davies R, ElGuindy A, Jefferson BK, Patel TN, Patel M, Chandwaney RH, Mastrodemos OC, Rangan BV, Brilakis ES. Prevalence and outcomes of balloon undilatable chronic total occlusions: Insights from the PROGRESS-CTO. Int J Cardiol. 2022 Sep 1;362:42-46. doi: 10.1016/j.ijcard.2022.04.057. Epub 2022 Apr 26. PMID: 35483480.)  are not mentioned in the manuscript and it is not clear what was the difference between the aims of the present work and the before one. The present analysis includes more cases, but the interval of the collection time is not specified against the before one (between 2012 and 2022). I suggest investigating the recent tendencies in the database according to the new results.

E.g. in the present manuscript, the numbers of the rotational (23%: 1994 cases) and the orbital atherectomies (2%: 173 cases) have increased significantly compared to the before data. It should be worth giving data about the rate of combined usage of the different types atherotomies. I am curious the opinion of the authors the preference and the strategy for applying the two methods.

 Response: We greatly appreciated the reviewer’s insightful comment. The prior study from Xenogiannis et al analyzed the use of atherectomy during chronic total occlusion (CTO) in the PROGRESS CTO registry between 2012 and 2018. In that study rotational atherectomy was used in 105 (2.9%) cases, orbital atherectomy in 8 (0.2%), and both in 4 (0.1%) cases. Furthermore, atherectomy was used in 51 cases (1.4%) bail out strategy for “balloon-uncrossable” or/and “balloon undilatable” lesions, while it was used for planned strategy “lesion preparation” in 66 (1.8%) cases. Our analyses focuses only on balloon uncrossable lesions, not the atherectomy cases for balloon undilatable lesions or for lesion preparation. The following section was added to the discussion:

‘In a prior study from the PROGRESS CTO registry examining the use of atherectomy during chronic total CTO PCI, atherectomy was used in 51 cases (1.4%) as bail out strategy for “balloon-uncrossable” or/and “balloon undilatable” lesions. The cases with “balloon uncrossable” and “balloon undilatable” lesions in which atherectomy was used had higher technical success rates (92% vs. 79%, p=0.032) and procedural (90% vs. 79%, p=0.046) success rates and compared with similar lesions not-treated with atherectomy. MACE rates were similar (7% vs. 4%, p=0.422). [26]’

The study by Simsek et al focuses on balloon undilatable lesions, while our study focuses on balloon uncrossable lesions. The study has been added to the discussion section “In a prior publication from the PROGRESS CTO registry the prevalence of balloon uncrossable lesions following successful guidewire crossing was 9.0% [2], whereas the prevalence of balloon uncrossable or balloon undilatable lesions was 15.5%, [10] and the prevalence of balloon undilatable lesions was 8.5%.[11]”

In our study rotational atherectomy was used in in 179 cases, 23% of the balloon uncrossable cases (overall balloon uncrossable: 795 cases), orbital atherectomy was used in 16 cases, 2% of the balloon uncrossable cases. The number of cases was clarified in the results section: “Several techniques were used in balloon uncrossable lesions, such as guide catheter extensions in 267 cases (34%), grenadoplasty in 198 cases (25%), rotational atherectomy in 179 cases (23%) and laser in 140 cases (18%, Figure 2). Rotational atherectomy combined with laser atherectomy was used in 31 cases (4%) while rotational atherectomy together with orbital atherectomy was used in 3 cases (0.4%).”

We added the following section to the discussion: “The combined use of Rotational Atherectomy and excimer LaSER is called the “RASER” technique, and consists of upfront use of laser atherectomy followed by rotational atherectomy in heavily calcified lesions after failure of Rotawire delivery.[3,21]” An advantage of laser atherectomy is that it can be performed over any standard 0.014 inch guidewire (but should be performed with caution over polymer-jacketed guidewires due to risk of “melting” the polymer). After successfully crossing further modification can be performed with balloons (non-compliant, high-pressure, scoring, cutting) atherectomy devices, or intravascular lithotripsy. [15,21]”

Reviewer #2

Well written paper, important clinical features for interventionalists including technical and safety endpoints well described.

3 Questions to authors:

1) do you have a defined number of use of intravascular lithotripsy since this method is available since 3-4 years

Response: Thank you for this insightful comment. We have added the following sentence to the results section: “Intravascular lithotripsy was used in 5 cases (0.6%).”

 2) do you have information about patients readmitted for repeat revasc or events at extend follow up?

Response: Thank you for the comment. Unfortunately we have very limited follow-up data in the PROGRESS CTO registry.

3) is there any information available concerning duration and composition of DAPT in these patients?

Response: We appreciated the reviewer’s comment, but unfortunately we do not collect information on duration and composition of DAPT in the PROGRESS CTO registry.

We believe that the reviewers’ comments and our response to those comments have significantly improved our manuscript and we look forward to your repeat review.

Sincerely,

Judit Karacsonyi, MD, PhD

Emmanouil S. Brilakis, MD, PhD

Reviewer 2 Report

Well written paper, important clinical features for interventionalists including technical and safety endpoints well described.

3 Questions to authors:

1) do you have a defined number of use of intravascular lithotripsy since this method is available since 3-4 years

2) do you have information about patients readmitted for repaet revasc or events at extend follow up?

3) is there any information available concerning duration and compsoition of DAPT in these patients?

3)

Author Response

(The authors gave the same response as above.)
